# Enhancing Aboveground Biomass Estimation for Three Pinus Forests in Yunnan, SW China, Using Landsat 8

**Jing Tang, Ying Liu, Lu Li, Yanfeng Liu, Yong Wu, Hui Xu and Guanglong Ou ***

Key Laboratory of State Forestry Administration on Biodiversity Conservation in Southwest China, Southwest Forestry University, Kunming 650224, China
* Correspondence: olg2007621@swfu.edu.cn

**Abstract:** The estimation of forest aboveground biomass (AGB) using Landsat 8 operational land imagery (OLI) images has been extensively studied, but forest aboveground biomass (AGB) is often difficult to estimate accurately, in part due to the multi-level structure of forests, the heterogeneity of stands, and the diversity of tree species. In this study, a habitat dataset describing the distribution environment of forests, Landsat 8 OLI image data of spectral reflectance information, as well as a combination of the two datasets were employed to estimate the AGB of the three common pine forests (*Pinus yunnanensis* forests, *Pinus densata* forests, and *Pinus kesiya* forests) in Yunnan Province using a parametric model, stepwise linear regression model (SLR), and a non-parametric model, such as random forest (RF) and support vector machine (SVM). Based on the results, the following conclusions can be drawn. (1) As compared with the parametric model (SLR), the non-parametric models (RF and SVM) have a better fitting performance for estimating the AGB of the three pine forests, especially in the AGB segment of 40 to 200 Mg/ha. The non-parametric model is more sensitive to the number of data samples. In the case of the *Pinus densata* forest with a sample size greater than 100, RF fitting provides better fitting performance than SVM fitting, and the SVM fitting model is better suited to the AGB estimation of the *Pinus yunnanensis* forest with a sample size of less than 100. (2) Landsat 8 OLI images exhibit superior accuracy in estimating the AGB of the three pine forests using a single dataset. Variables, such as texture and vegetation index variables, which can reflect the comprehensive reflection information of ground objects, play a significant role in estimating AGBs, especially the texture variables. (3) By incorporating the combined dataset with characteristics of tree species distribution and ground object reflectance spectrum, the accuracy and stability of AGB estimation of the three pine forests can be improved. Moreover, the employment of a combined dataset is also effective in reducing the number of estimation errors in cases with AGB less than 100 Mg/ha or exceeding 150 Mg/ha.

**Keywords:** aboveground biomass; habitat dataset; Landsat 8-OLI images; pine forest; model comparison

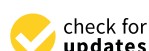



## 1. Introduction

In addition to regulating the water supplies and climate, forests accumulate biomass by absorbing light, water, carbon dioxide, and other compounds [1]. Forests are the largest carbon source in terrestrial ecosystems [2], accounting for more than two-thirds of the total carbon sequestration annually [3,4]. Therefore, it is crucial to accurately estimate forest biomass to thoroughly understand the carbon cycle and carbon balance in terrestrial ecosystems [5].

There are two types of forest biomass: aboveground biomass (AGB) and underground biomass (UB) [6]. Because AGB accounts for 70% to 90% of the total forest biomass [7], most previous studies have focused on AGB. The estimation methods of AGB include destructive and non-destructive methods. The destructive harvest method is considered to be the most accurate AGB estimation method. However, as this method poses a threat to

the flora and fauna, it is inappropriate for large-scale AGB estimation [8]. Due to the multi-scale and multi-band characteristics of remote-sensing (RS) technology, it can effectively ensure the spatial integrity and temporal continuity of data. RS has become a common method for large-scale non-destructive AGB estimation in recent decades [9]. Optical remote sensing was developed first, and a large number of data sources have been used to estimate thousands of forest AGBs [6,10–13]. The natural forest consisting of mature stands, however, is characterized by a multi-layer canopy and high density, resulting in the saturation of spectral reflectance in optical remote sensing and, in turn, leading to an underestimation of large AGB values [1]. The AGB of young forests, on the other hand, will be overestimated due to sparse canopy radiation combined with understory vegetation, soil, and other radiation information. These are the main difficulties that have arisen in estimating forest AGB using an optical RS dataset. As a result of their capability of obtaining detailed information on forest structure, radar and LiDAR have become two of the main techniques for forest AGB estimation in recent decades [13–21]. However, the number of data sources of radar and LiDAR is relatively smaller than the number of optical RS datasets; the working mechanism of these data sources also limits their application [6]. Thus, optical RS still remains an effective data source for AGB estimation in a wide area. As a result, Landsat images have been widely used in AGB estimation, in part due to their accessibility, time continuity, and moderate spatial resolution [22–25]. It should be noted that in forest AGB estimation using Landsat 8 OLI images, the problem of underestimating high values and overestimating low values when using this data source is still challenging [26].

In order to overcome the uncertainty that can be caused by using only one remote-sensing source in the estimation of forest AGBs, researchers have become increasingly aware of the necessity of multi-source data to improve the accuracy of biomass estimation in forests [6]. For example, by combining radar or UAV data with optical remote-sensing data, it is possible to overcome the challenge of AGB estimation caused by changes in the forest structure and the high heterogeneity of the forest landscapes [22,23,26–28]. The complementarity between remote-sensing data can improve the estimation accuracy of AGB to a certain extent. However, since radar and UAV data sources are limited, the combination of non-remote-sensing datasets, such as field measurement and forest surveys, with optical remote sensing can also contribute to the improvement of AGB estimation accuracy [29–33]. In recent years, some scholars have emphasized the fact that trees have a long growing season. Forest AGB estimations can be enhanced by incorporating long-term data, such as climate [34,35] and phenology [36–38], into an optical remote-sensing dataset to compensate for the lack of timeliness in remote-sensing data for forest AGB estimations.

Habitat [39] is a concept that was first proposed by Grinnel in 1917. In general, it refers to the space in which an individual, a population, or a community can complete their life processes [40]. In many studies, habitat has been shown to affect biomass accumulation and plant allocation, as well as changes in vegetation genotypes [41–44]. While adapting to the environment, vegetation also has varying carbon sequestration abilities due to the accumulation and differentiation of biomass in different parts of the foliage, resulting in the change of habitat [45]. There is a close relationship between species and habitat [46]. This relationship exists due to the statistical association of habitat information with species abundance or the probability of occurrence [47]. The abundance of species in a region is influenced by the community structure at different scales [48], and forests are no exception. Studies have confirmed that the complexity of the vertical structure of forests is related to forest biodiversity [49,50]. Thus, the habitat information that can indirectly measure the biodiversity information in the region [51] can represent the vertical structure information of the forest to a certain extent, especially on a large spatial scale [52].

Typical habitat data characteristics include climate, slope, aspect, elevation, soil type, etc. The habitat suitability value (HSV) is determined by the comprehensive effects of different habitat factors on species in a region. In ecology and conservation biology, species distribution maps are often used to indicate the HSV of species in specific areas [53,54]. However, it is difficult to produce a map of the species distribution for all species. It is

especially difficult to obtain the HSV of most species in areas where there are abundant species [55]. Therefore, it is an effective method to import habitat data into species distribution models (SDMs) to obtain the HSV of species. A number of common SDMs include BIOCLIM, ENFA, HABITAT, Maxent, etc. Maxent is one of the most widely used and highly accurate SDMs [56] and has been extensively used in species conservation and management [57–59]. This model calculates the binding force of species distribution according to the environmental factors of known distribution points; then, it estimates the probability of species distribution in unknown distribution points [60] and ultimately obtains the HSV of the species. However, the number of studies that use a habitat dataset as an independent variable to estimate forest AGB is still too small [38], especially for pine forest AGBs.

Furthermore, in addition to the data used for estimation, AGB estimation is also strongly influenced by the choice of modeling algorithms [61]. There are two types of AGB modeling algorithms: parametric and non-parametric. The parametric model determines the relationship between AGB and independent variables through linear functions, power functions, exponential functions, etc. Since this model requires fewer sample data and can quantify the relationship between AGB and the variables [1], it has become the most commonly utilized biomass modeling algorithm [6]. Due to the complex correlation between the variables and AGB, the parametric model cannot provide an adequate estimation [61,62]. However, the stepwise linear regression algorithm can differentiate between the important variables in the modeling through the significance test, which, in turn, improves the estimation performance of the parametric model to a certain extent [63]. In addition, non-parametric models are also capable of handling nonlinear relationships and can be used to determine the most suitable model structure for the dependent variable from the estimated data, which has been widely used in forest AGB estimation in the past ten years [8,25,33,64]. Random forest and support vector machines are two of the most commonly used non-parametric models. However, the non-parametric model has two main shortcomings; the first is that it is sensitive to data [1,65], and the second is that its interpretation for the estimation process is less clear than that of the parametric model [6].

In this study, the AGB of the three common pine forests (*Pinus yunnanensis* forests, *Pinus densata* forests, and *Pinus kesiya* forests) was estimated in Yunnan Province, southwest China, by employing varying datasets (e.g., a Landsat 8 OLI dataset, a habitat dataset, and a combined dataset composed of them both) and modeling algorithms (e.g., stepwise linear regression (SLR), random forest (RF), and support vector machine (SVM)). Further, the performance of AGB estimations of different forests under different models was compared.

The purpose of this study was to answer the following questions:

(1) Do estimation models have an impact on the AGB estimation for pine forests?
(2) Is it possible to estimate the AGB for the three pine forests using the habitat dataset?
(3) Does the employment of a habitat dataset reduce the probability of overestimation and underestimation of the AGB estimation?

## 2. Materials and Methods

The AGB estimations of pine forests included in this study were carried out using different datasets from the following steps: (1) the selection of sample plots and calculation of AGBs; (2) habitat simulation of pine forests in order to obtain habitat datasets; (3) preprocessing of Landsat 8 OLI images in order to obtain RS datasets; (4) analysis of the correlation between habitat datasets and RS datasets and plot AGBs in order to select independent variables; (5) combining data from selected habitat data and RS data; (6) the development of AGB estimation models (e.g., SLR, RF, and SVM) using different datasets; (7) comparing the modeling results of AGB estimations. The specific process is shown in Figure 1.

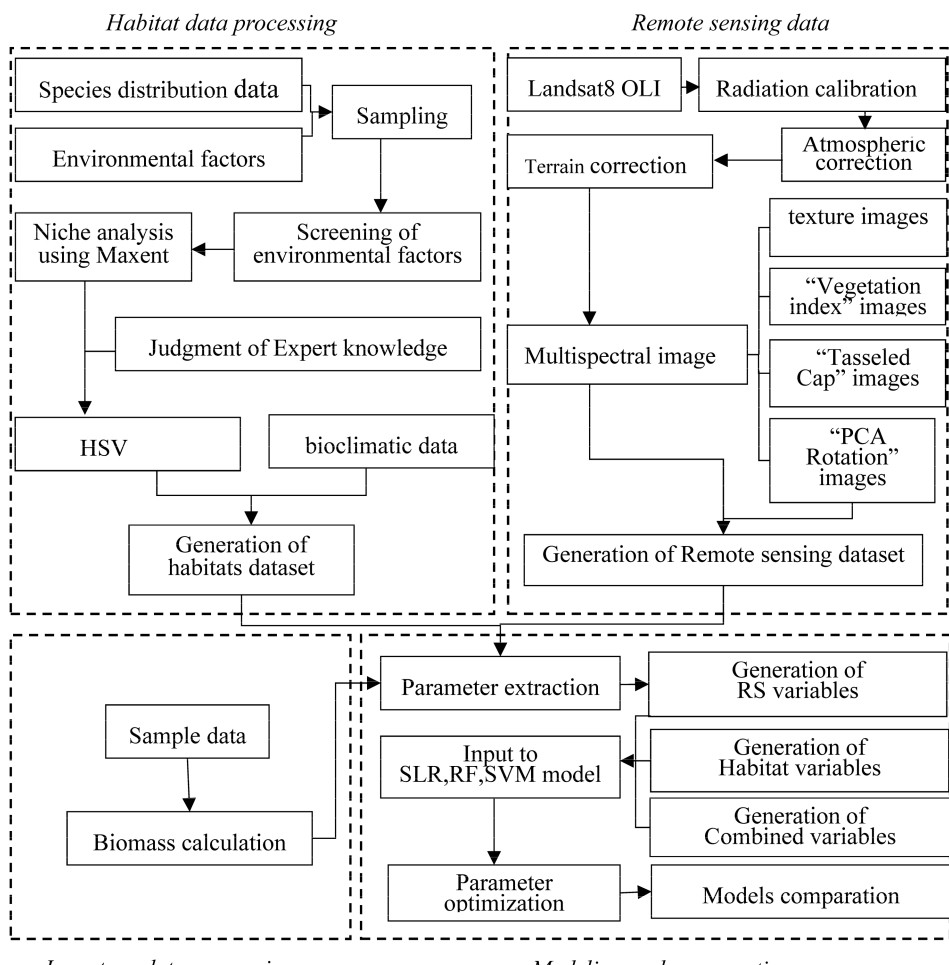

**Figure 1.** Flow chart of estimating aboveground biomass (AGB) of pine forest using different datasets through parametric model and non-parametric model (Note: HSV, habitat suitability value; SLR, stepwise linear regression; RF, random forest; SVM, support vector machine).

### 2.1. Study Area

Yunnan is located in southwest China. It is one of the three forest regions in China. The pine forests in the province mainly comprise *Pinus yunnanensis*, *Pinus densata*, and *Pinus kesiya* trees [30,65,66]. The forests play an important role in ecological services and forest carbon sinks in the region [33,67], and forest biomass is the basis for forest carbon sink estimation [68]. The study areas of this study were selected in Yongren County, Shangri-La City, and Pu'er City, where the main distribution areas of the three pine forests are located (Figure 2).

Yongren County is located in the central and northern parts of Yunnan Province. The altitude of this area ranges from 1530 to 1700 m above sea level, and the terrain is relatively flat. The climate of this region is classified as a north subtropical southwest monsoon, and the precipitation season lasts from June to October, when the dry and wet seasons are clearly distinguishable [69]. There are mainly two types of vegetation in this region: sub-humid subtropical evergreen broad-leaved forests and *Pinus yunnanensis* secondary forests. *Pinus yunnanensis* grows at an altitude of 1000–3200 m and can withstand drought as well as barren soil. This tree species is native to this area, and it is one of the pioneer tree species in southwest China. The Baima River Forest Farm in Yongren County is the largest mother forest base for *Pinus yunnanensis* in China.

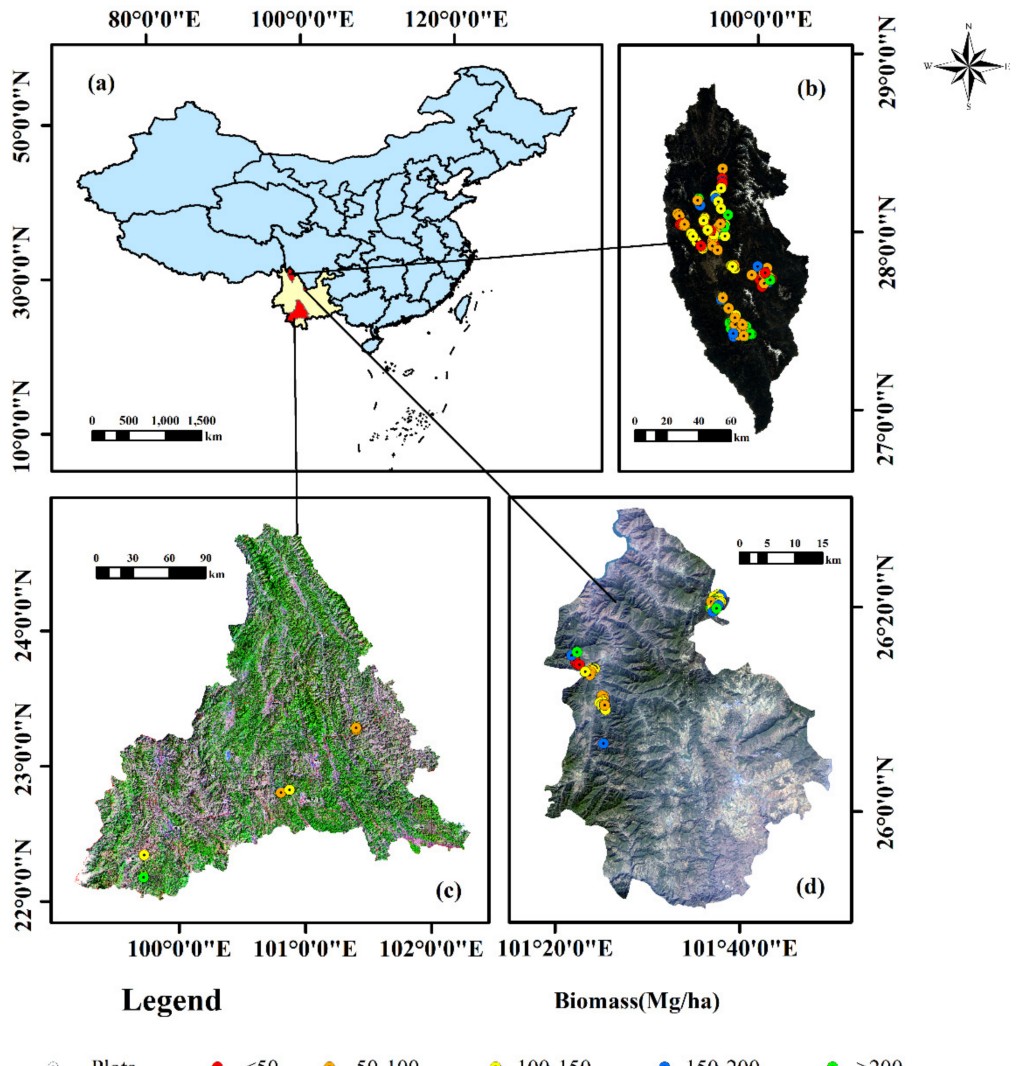

**Figure 2.** (**a**) Location of the study area; (**b**) RGB true color composite image of Shangri-la City and field plots of *Pinus densata* forest; (**c**) RGB true color composite image of Pu'er City and field plots of *Pinus kesiya* forest; (**d**) RGB true color composite image of Yongren County and field plots of *Pinus yunnanensis* forest.

Shangri-la, with an altitude of between 1503 and 5545 m above sea level, is located in the northwest of Yunnan Province. The annual average temperature in this area is 5.5 °C, and summer and autumn are the seasons with the most precipitation. This area is part of the World Natural Heritage site, "Parallel Flow of the Three Rivers", with rich forest resources. *Pinus yunnanensis*, *Pinus densata*, *Picea* spp, *Abies* spp, and *Quercus* spp are the dominant tree species in the Shangri-la region [70]. Among the tree species, *Pinus densata* is considered to be a unique tree species in the alpine region of western China, and its vertical distribution is slightly higher than that of *Pinus yunnanensis*.

Pu'er is located in the southwest of Yunnan Province, where the altitude ranges from 376 to 3306 m above sea level. In addition, more than 90% of the area is mountainous. In most parts of this area, there is no frost all throughout the year, and the annual precipitation is between 1100 and 2780 mm. In addition, its climate is classified as the plateau monsoon climate of the south Asian tropics. The abundant precipitation keeps the relative humidity in the region at 82% all throughout the year, resulting in favorable conditions for vegetation growth [71]. Pu'er is the second largest forest region in the Yunnan Province. Over half of the area is covered by *Pinus kesiya*, the dominant tree species. *Pinus kesiya* is also a main

afforestation tree species that can be found at altitudes below 1800 m in southern, central, and western Yunnan province.

### 2.2. Sample Plot Data and Forest AGB

In order to obtain the AGB of the pine, destructive samples were taken from 87 *P. yunnanensis* plots, 147 *P. densata* plots, and 45 *P. kesiya* plots in the study area from 2011 to 2017. The sample plots of 30 m × 30 m were selected and established according to the stock state map based on tree age, altitude, slope, aspect, and sampling distance of 1 km. The coordinates, altitude, slope, aspect, DBH, and tree height of the sample plots were recorded in the process of sampling.

Since there is no calculation model for these three AGB of pine trees in the existing biomass model, in this study, 3 to 5 stands with the average level of stands in the sample plot were selected for cutting, and subsequently, the biomass of their trunks, bark, branches, and leaves in the aboveground part was measured. The biomass of the wood and the bark of the trees was measured by taking a 3 cm disk along the trunk of each tree at 2 m intervals, and then, the density of the samples was measured using the drainage method. The disk was dried in an oven at a constant temperature of 105 °C, and then, the biomass of the disk was determined by comparing the weight before and after drying. In addition, the volume of wood and bark from each sample tree was converted into biomass in 2 m units according to the density of the samples. By using similar methods, the branches and leaves were collected and weighed by grade, and the resulting biomass was measured accordingly. The total AGB of each sample was obtained by adding the AGB from different parts of the sampled trees. Finally, the power function was used to fit individual tree AGB data. The AGB fitting formulae of *Pinus yunnanensis* [65], *Pinus densata* [1], and *Pinus kesiya* [66] were obtained, as shown in Formula (1)–(3).

$$AGB_i = 0.048 * DBH^{1.9276} * H^{0.9638} \tag{1}$$

$$AGB_i = 0.073 * DBH^{1.739} * H^{0.880} \tag{2}$$

$$AGB_i = 0.058 * DBH^{2.12} * H^{0.4668} \tag{3}$$

In these formulae, DBH (cm) is the average diameter at breast height (1.3 m), H (m) is the average tree height, and $AGB_i$ is the aboveground biomass of a single standing tree (kg).

In order to obtain the AGB of the sample plot, the unit was converted into the value per hectare using equation (4). The final AGB statistical data of the three pine forests are shown in Table 1.

$$AGB_p = \frac{AGB_i \times n}{30 \times 30} \times \frac{10,000}{1000} \tag{4}$$

**Table 1.** The statistical parameters of AGB in the sample plot.

| Species | Number of Plots | Statistical Indicators | AGB (Mg/ha) |
|---|---|---|---|
| *Pinus yunnanensis* | 87 | Min. | 17.901 |
| | | Max. | 287.679 |
| | | Mean | 114.868 |
| *Pinus densata* | 147 | Min. | 2.114 |
| | | Max. | 344.382 |
| | | Mean | 121.474 |
| *Pinus kesiya* | 45 | Min. | 49.063 |
| | | Max. | 204.448 |
| | | Mean | 116.432 |

In this formula, $AGB_i$ is the biomass of individual trees, n is the number of trees in the sample plot, and $AGB_p$ is the AGB of the sample plot (Mg/ha).

### 2.3. Acquisition of Remote-Sensing Datasets

The Landsat 8 OLI images and DEMs employed in this study were downloaded from http://www.gscloud.cn/ (accessed on 6 August 2022). The spatial resolution of these data was 30 m, and the coordinate system was a Universal Transverse Mercator with zone 47 north as the spatial reference frame. The specific parameters of the Landsat 8 OLI images are shown in Table 2. Radiometric calibration, FLASSH atmospheric correction, and C-correction topographic correction were performed in ENVI in order to correct the radiometric errors in the images. Subsequently, the corrected images were mosaicked and clipped to obtain the Landsat 8 OLI image of the study area.

**Table 2.** The specific parameters of Landsat 8 OLI images.

| Study Area | Image ID | Average Cloud Cover (%) | Start Time |
| --- | --- | --- | --- |
| Yongren | LC81300422016030LGN00 | 0.00 | 30 January 2016 |
| Shangri-la | LC81310412016325LGN00 | 0.40 | 20 November 2016 |
| | LC81320402016348LGN00 | 0.73 | 13 December 2016 |
| | LC81320412016348LGN00 | 0.76 | 13 December 2016 |
| Pu'er | LC81290442015052LGN00 | 0.08 | 21 February 2015 |
| | LC81290452015052LGN00 | 1.87 | 21 February 2015 |
| | LC81310432015066LGN00 | 0.00 | 7 March 2015 |
| | LC81310442015066LGN00 | 0.00 | 7 March 2015 |
| | LC81300432015075LGN00 | 0.18 | 16 March 2015 |
| | LC81300442016046LGN00 | 0.00 | 15 February 2016 |
| | LC81300452016046LGN00 | 0.01 | 15 February 2016 |
| | LC81310452016069LGN00 | 0.41 | 9 March 2016 |

The corrected Landsat 8 OLI image includes seven multi-spectral bands, namely Band1-Coastal, Band2-Blue, Band3-Green, Band4-Red, Band5-NIR, Band6-SWIR1, and Band7-SWIR2. In this study, for the purpose of obtaining more data for the forest AGB estimations, sixty-four conversion variables from remote-sensing images were also employed. A total of five commonly used vegetation indices were calculated, namely the normalized difference vegetation index (NDVI), simple vegetation index (SVI), enhanced vegetation index (EVI), atmospherically resistant vegetation index (ARVI), and structurally insensitive pigment index (SIPI), as well as three tasseled cap images, namely brightness, greenness, and humidity. In addition, the first, second, and third principal component data, which can reflect more than 75% of the image information, were also calculated by ENVI 5.3 software. Additionally, a total of 56 Landsat 8 OLI texture variables were also calculated under $3*3$ Windows based on the gray co-occurrence matrix, including homogeneity, anisotropy, mean, angle second moment, entropy, correlation, variance, and contrast. Subsequently, a Pearson correlation analysis was conducted to screen the AGB-related candidates that passed the significance test ($p \leq 0.01$) as the remote-sensing data. Finally, the RS dataset was derived from the resulting remote-sensing data. The selected remote-sensing variables and their correlation with AGB are shown in Figure 3. In the figure, the variables superscripted with "-" are factors that show a negative correlation with AGB.

### 2.4. Acquisition of Habitat Datasets

When the species adapt to light and precipitation after a prolonged period of time, they will form development rules that will increase their survival advantages in the habitat conditions. The 19 bioclimatic variables downloaded from WorldClim (https://www.worldclim.org/, accessed on 6 August 2022) are considered to be the most significant habitat factors because they can reflect temperature, precipitation, and some other aspects. Over 85% of habitat studies use these 19 bioclimatic factors [72], which are included in Table 3. In this study, the HSV of pine forests and 19 bioclimatic environmental variables were employed as habitat candidates to estimate the AGB.

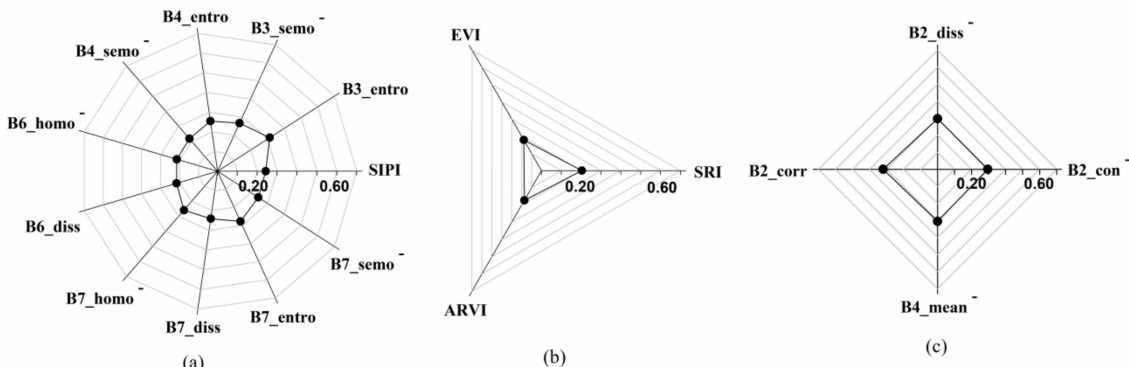

**Figure 3.** Radar plot of RS data associated with forest AGB (**a**) *Pinus yunnanensis* forests; (**b**) *Pinus densata* forests; (**c**) *Pinus kesiya* forests.

**Table 3.** The 19 bioclimatic environmental variables from WorldClim.

| Variable Code | Variable Description | Variable Code | Variable Description |
|---|---|---|---|
| Bio1 | Annual mean temperature | Bio2 | Mean diurnal range |
| Bio3 | Isothermality | Bio4 | Temperature seasonality |
| Bio5 | Max temperature of the warmest month | Bio6 | Min temperature of the coldest month |
| Bio7 | Range of annual temperature | Bio8 | Mean temperature of the wettest quarter |
| Bio9 | Mean temperature of the driest quarter | Bio10 | Mean temperature of the warmest quarter |
| Bio11 | Mean temperature of the coldest quarter | Bio12 | Annual average precipitation |
| Bio13 | Precipitation of the wettest month | Bio14 | Precipitation of the driest month |
| Bio15 | Precipitation seasonality | Bio16 | Precipitation of the wettest quarter |
| Bio17 | Precipitation of the driest quarter | Bio18 | Precipitation of the warmest quarter |
| Bio19 | Precipitation of the coldest quarter | | |

In order to avoid data disaster in subsequent analyses, a Pearson correlation analysis was conducted among the 19 bioclimatic environmental variables in SPSS 22.0 for Windows to select variables with a correlation coefficient absolute value lower than 0.9 and variables with significance to the pine habitat. The habitat constraint factors of *Pinus yunnanensis* forests included 10 variables, namely Bio1, Bio3, Bio4, Bio5, Bio6, Bio8, Bio10, Bio13, Bio14, Bio15, and Bio16, while the variables of *Pinus densata* forests were Bio2, Bio3, Bio4, Bio5, Bio6, Bio7, Bio14, Bio15, and Bio16. Furthermore, the significant variables for the growth of *Pinus kesiya* forests included Bio1, Bio2, Bio4, Bio6, Bio7, Bio10, Bio13, Bio17, and Bio18.

Species distribution point data were obtained from the Chinese Virtual Herbarium (https://www.cvh.ac.cn/, accessed on 6 August 2022) and the Global Biodiversity Information Facility (https://www.gbif.org, accessed on 6 August 2022). However, these downloaded points were difficult for unifying the collection time and the collector, and the data description was not standard. Therefore, it was necessary to screen these points and delete duplicate points and questionable points to ensure the accuracy of the data. Finally, 803 sample points were obtained, including 460 *Pinus yunnanensis*, 319 *Pinus densata,* and 24 *Pinus kesiya*.

The selected bioclimatic variables and species distribution points were imported into Maxent, while 25% of them were set as random test points, and the model was repeatedly applied ten times in order to obtain the HSV of the three pine forests located in southwest China. The anticipated results were evaluated using the methods of the receiver operating characteristic curve (ROC) and Jackknife. The results showed that the habitat fitting AUC values of the *Pinus yunnanensis* forest, *Pinus densata* forest, and *Pinus kesiya* forest were 0.9889, 0.9906, and 0.9983, respectively. It is generally accepted that the predicted result is accurate when the AUC value is greater than 0.9 [73,74]. The habitat simulation AUC values for all three pine forests were higher than 0.98. Additionally, the fitted AUC standard deviation of the *Yunnan pine* forest and *Pinus densata* forest was 0.0018, and *Pinus kesiya* forest was 0.0004, all of which were less than 0.002. The accuracy was guaranteed. By

consulting with experts, the HSV variable values that were closest to the actual species distribution were selected, as expert knowledge helped improve Maxent's prediction accuracy [55].

The candidate of habitat data was derived from the HSV of the pine forests and the 19 bioclimatic variables. Subsequently, a Pearson correlation analysis was conducted to determine the AGB-related candidates that passed the significance test ($p \leq 0.01$) as the habitat data. Finally, the habitat dataset was derived from the resulting habitat data. The selected habitat variables and their correlation with AGB are shown in Figure 4. In the figure, the variables superscripted with "-" are factors that show a negative correlation with AGB.

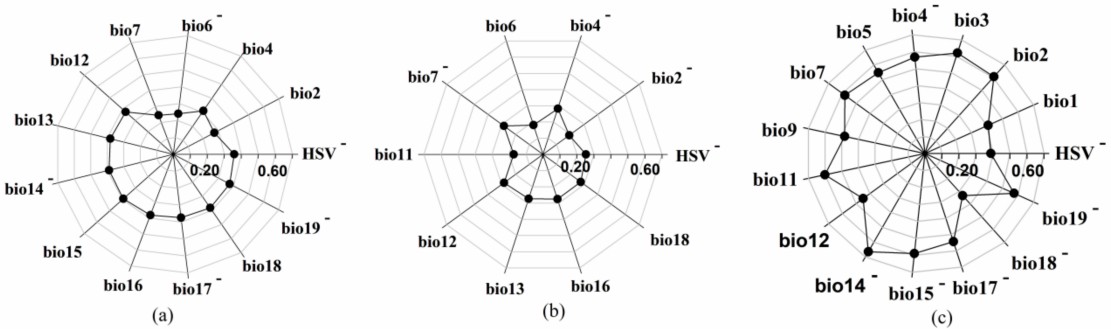

**Figure 4.** Radar plot of habitat data associated with forest AGB (**a**) *Pinus yunnanensis* forests; (**b**) *Pinus densata* forests; (**c**) *Pinus kesiya* forests.

### 2.5. Acquisition of Combined Datasets

The combined dataset employed for the AGB estimation consisted of a habitat dataset and an RS dataset, which correlated with the AGB (correlation coefficient > 0.1) and passed the significance test ($p \leq 0.01$).

The combined dataset of *Pinus yunnanensis* forests consisted of 13 habitat data and 11 remote-sensing data; the combined dataset of *Pinus densata* forests consisted of 10 habitat data and 3 remote-sensing data; and finally, the combined dataset of *Pinus kesiya* forests consisted of 15 habitat data and 4 remote-sensing data.

### 2.6. AGB Modeling Algorithms

In this study, a parametric model, SLR, and two additional non-parametric models, namely RF and SVM, were used for AGB modeling based on a different dataset.

#### 2.6.1. Stepwise Linear Regression (SLR)

The stepwise linear regression model builds a prediction model by first calculating the significance of variables and then deleting the variables with low significance backward or adding the variables with significance forward to the prediction model. Thereby, it can effectively solve the collinearity problem between explanatory variables [75]. The constructed model can be expressed using Formula (5).

$$Y = b_0 + b_1 X_1 + b_2 X_2 + \cdots + b_n X_n \tag{5}$$

In this formula, $b_1$, $b_2$, ..., $b_n$ are the regression coefficients of the prediction variables, and $b_0$ is the constant of the prediction model. The stepwise backward linear regression model was utilized in this study.

#### 2.6.2. Random Forest (RF)

A random forest is a machine-learning algorithm used for classification and regression. Its basic idea is to generate a new training sample set by repeatedly performing random extractions on two-thirds of the data from the original training set, and the remaining data that are not extracted become the out-of-bag detection data. N regression decision trees

were constructed for the newly generated sample set to fully grow into a random forest. Finally, the best regression result was selected by voting on the prediction results of the decision tree.

By employing the multi-branch combinative learning method of random forests, it is possible to avoid the shortcomings of using only one classification [76], and the method has a good tolerance for outliers and noise in large-scale datasets [53]. Random forest has also been widely used in many other fields, such as agriculture and forestry, in recent years [77–79].

### 2.6.3. Support Vector Machine (SVM)

The SVM is a machine-learning method based on the theory of small sample statistics [64]. Its two main functions are, first, to find a hyperplane that fits the test data to the best degree, and second, to perform a two-dimensional segmentation to maximize the isolation edge of the data on both sides of the hyperplane in order to ensure the classification accuracy of the data [80]. This method can be utilized to effectively solve the problems of nonlinear data and high-dimensional pattern recognition [81]. The SVM model has a wide range of applications in image recognition [82], time series prediction [83], and so on.

In order to simulate the relationship between AGB and the estimation variables of pine forests, the "MASS", "randomForest", and "e1071" packages in R software were employed to construct the models of SLR, RF, and SVM.

### 2.7. Model Evaluation

An RS dataset, a habitat dataset, and a combined dataset were applied to all models of AGB estimation. In this study, the coefficient of determination ($R^2$), the root mean square error (RMSE), and the normalized root mean square error (NRMSE) were utilized to evaluate the accuracy of model fitting. Furthermore, the mean error (ME), mean relative error (MRE), and mean absolute relative error (MARE) were utilized to measure the overall prediction accuracy and the segment prediction accuracy with a 50 Mg/ha interval (<50, 50–100, 100–150, 150–200, >200 Mg/ha).

$$R^2 = 1 - \frac{\sum\limits_{i=1}^{n} \left( \hat{y}_i - y_i \right)^2}{\sum\limits_{i=1}^{n} \left( y_i - \overline{y} \right)^2} \tag{6}$$

$$RMSE = \sqrt{\frac{\sum\limits_{i=1}^{n} \left( y_i - \hat{y}_i \right)^2}{n}} \tag{7}$$

$$NRMSE = \frac{RMSE}{\overline{y}} \tag{8}$$

$$ME = \frac{\sum_{i=1}^{n} \left( y_i - \hat{y}_i \right)}{n} \tag{9}$$

$$NRMSE = \frac{RMSE}{\overline{y}} \tag{10}$$

$$MARE = \frac{\sum\limits_{i=1}^{n} \left| \frac{y_i - \hat{y}_i}{\hat{y}_i} \right|}{n} \times 100\% \tag{11}$$

In this table, $\hat{y}_i$ and $y_i$ are the predicted AGB and the corresponding AGB in the sample plot; $\overline{y}$ is the mean AGB of the sample plots; and n is the number of samples.

## 3. Results

To compare the role of the model and the dataset in AGB estimation, the sample data were randomly divided into two groups: 70% of the sample data were fitting data for model construction, and the remaining 30% were testing data for model validation. Due to the differences in the number of sample plots in pine forests, some appropriate adjustments were made to the datasets of different pine forests. The aforementioned processing was performed in order to ensure the validity of model fitting and validation. The final statistical values of the modeling and testing samples are shown in Table 4.

**Table 4.** Statistics of sample plot data used in this research.

| Species | Fitting | | | | Testing | | | |
|---|---|---|---|---|---|---|---|---|
| | Number | AGB Range (Mg/ha) | AGB Mean (Mg/ha) | AGB Std. Dev. (Mg/ha) | Number | AGB Range (Mg/ha) | AGB Mean (Mg/ha) | AGB Std. Dev. (Mg/ha) |
| *Pinus yunnanensis* | 57 | 17.9–287.7 | 115.1 | 56.9 | 30 | 40.6–270.2 | 114.4 | 53.1 |
| *Pinus densata* | 117 | 2.1–344.4 | 119.3 | 70.6 | 30 | 11.1–344.4 | 107.6 | 76.3 |
| *Pinus kesiya* | 30 | 49.1–204.4 | 116.2 | 40 | 15 | 70.1–192.2 | 116.8 | 33.6 |

### 3.1. Model Performance

AGB fitting was performed on the SLR, RF, and SVM of the three pine forests using the habitat dataset, RS dataset, and combined dataset, respectively. The $R^2$ by the SLR model for the AGB estimation of *Pinus yunnanensis* forests ranged from 0.1039 to 0.2514; the $R^2$ value of *Pinus densata* forests ranged from 0.0742 to 0.1650, and that of *Pinus kesiya* forests ranged from 0.0872 to 0.5331. The $R^2$ of the SLR was primarily below 0.6. When an RF model was implemented, the resulting $R^2$ of AGB fitting of *Pinus yunnanensis* forests ranged from 0.2028 to 0.7268, while *Pinus densata* forests ranged from 0.1903 to 0.7511, and *Pinus kesiya* forests ranged from 0.4617 to 0.8316. The $R^2$ distribution was mostly higher than 0.7. The $R^2$ of the SVM for the AGB fitting of the pine forest was mostly higher than 0.5. The $R^2$ of the *Pinus yunnanensis* forests ranged from 0.1791 to 0.8100; the *Pinus densata* forests ranged from 0.1559 to 0.7285; and the *Pinus kesiya* forest was concentrated between 0.5034 and 0.7956.

To further analyze the impact of different algorithms on AGB estimation, a boxplot of the three algorithms (SLR, RF, SVM) for AGB estimation was constructed, as shown in Figure 5. It can be seen that the median fitting coefficients of SLR, RF, and SVM were 0.1650, 0.7286, and 0.5361, respectively. The RF was significantly higher than the other two models, and the interquartile range (IQR) of RF was smaller than that of SVM. The median NRMSE of the three algorithms was 0.4350, 0.2634, and 0.2409, respectively. SLR had the largest error value, and RF was slightly higher than SVM. The IQR of RF was the smallest among the three models, and the RF model had the smallest estimation error dispersion. RF had the best performance of AGB estimation in the fitting data, followed by SVM and SLR, indicating that the non-parametric model had better AGB estimation characteristics for pine forests.

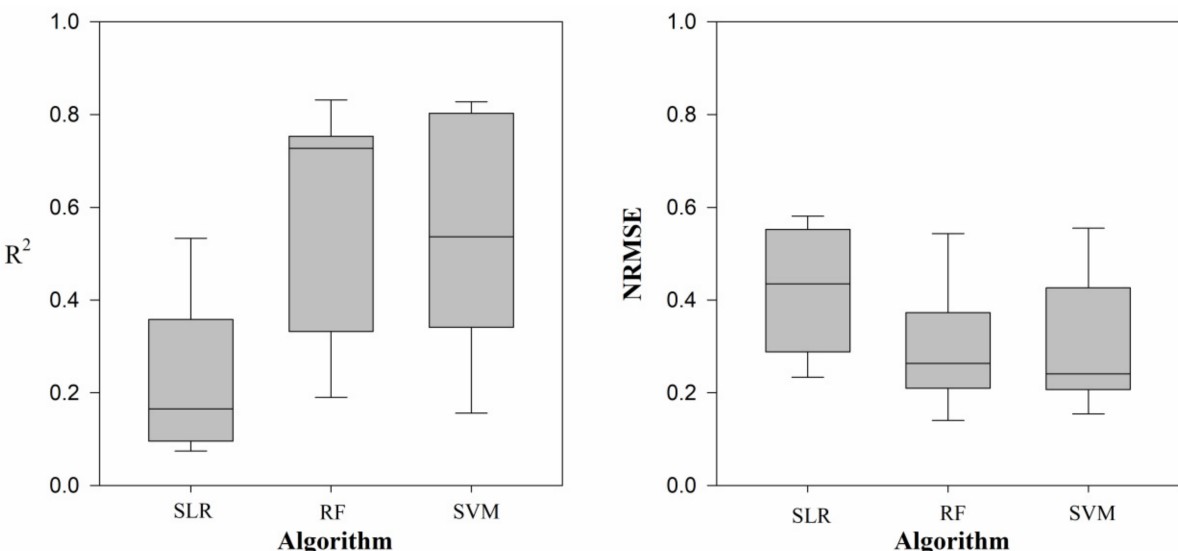

**Figure 5.** Boxplot of three algorithms for AGB estimation of the *Pinus* forest.

According to the results of a certain model, the dataset that provided the highest $R^2$ in the fitting data was applied to the AGB estimation of the test data; that is, the combined dataset was used for all, except for the RF, estimations of the *Pinus densata* forests and the SVM estimation of the *Pinus yunnanensis*, for which the RS dataset was used. Thereby, the errors of the varying forests were determined. Additionally, the independence test index of the model (Table 5) was represented by the mean value of each individual model error measurement index. It is evident that the fitting performance of AGB estimation in the test data was similar to that of the fitting data. Among the three models in question, RF had the lowest score of ME, MRE, and MARE. It was also found that the ME and MRE of the SLR were smaller than the predicted values of the SVM. However, the MARE, which is measured by the absolute value of the error, was higher than that of the SVM.

**Table 5.** Fitting and testing statistics of the three models.

| Model | Fitting | | | Testing | | |
|---|---|---|---|---|---|---|
| | $R^2$ | RMSE (Mg/ha) | NRMSE | ME (Mg/ha) | MRE (%) | MARE (%) |
| SLR | 0.3165 | 47.1631 | 0.4035 | −3.0224 | 3.2818 | 39.8129 |
| RF | 0.7698 | 27.1188 | 0.2320 | −1.8477 | −1.8103 | 30.5218 |
| SVM | 0.7840 | 26.1543 | 0.2238 | −4.5969 | −3.9566 | 35.5108 |

Furthermore, the prediction performance of the models was explained by the scatter diagram (Figure 6), which was formed by utilizing the estimated value of the model and reference data. It can be seen that the fitting performance of the non-parametric models (RF and SVM) was higher than that of the parametric models (SLR). SLR had estimation errors in all AGB segments, while RF and SVM had a higher estimation accuracy in cases with 40~200 Mg/ha, but in cases with AGB < 40 Mg/ha, the estimated value was significantly higher than the actual value; and in cases with AGB > 200 Mg/ha, the estimated value was significantly lower than the actual value. Although the SVM had a good fitting performance on some data, the estimation error in other cases was considerable. Compared with the SVM, the scatter points of the RF of *Pinus yunnanensis* forests and *Pinus densata* forests were distributed within a certain range around the complete fitting line. The distribution range of the SVM was larger than that of the RF, as was expressed in Figure 6. This shows that the overall estimation performance of the RF model was better than that of the SVM. However, in the case of *Pinus kesiya* forests, the scatter distribution of the RF and SVM models was similar.

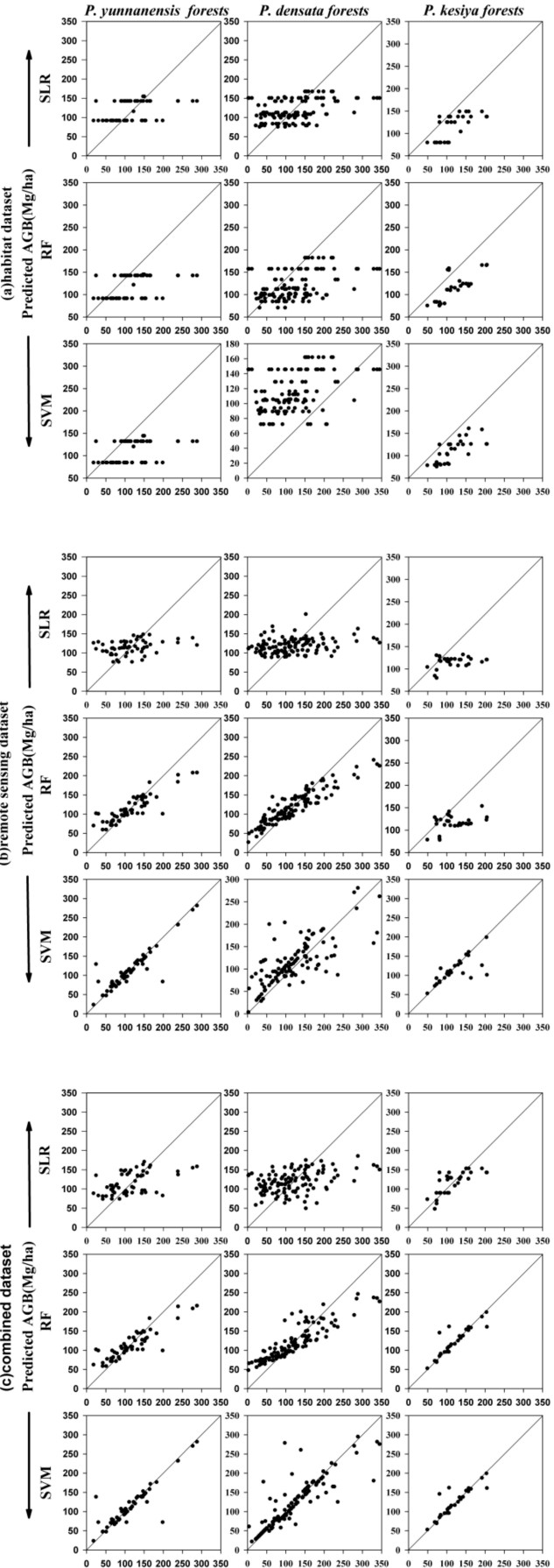

**Figure 6.** The predicted results of the SLR, RF, and SVM models of the different forests.

### 3.2. AGB Estimation Based on Different Datasets

In order to explain how the AGB estimation of pine forests is affected by a dataset, an RF with a higher accuracy performance for an AGB estimation was selected to perform AGB estimation of a habitat dataset, an RS dataset, and a combined dataset of them both.

It is evident from Table 6 that the AGB estimation performance of *Pinus yunnanensis* forests that utilized a different dataset under an RF, which was organized from high to low, was the combined dataset, the RS dataset, and the habitat dataset, respectively. In the case of *Pinus densata* forests, the habitat dataset had the lowest fitting performance. Although the fitting performance of the RS dataset was higher than the combined dataset, the error rate of the test data was also higher than the combined dataset. In the case of *Pinus kesiya* forests, the AGB estimation performance using the combined dataset was the highest, followed by the habitat dataset. However, the test error of the habitat dataset was higher than the RS dataset. On the whole, the combined dataset consisting of a habitat dataset and an RS dataset under RF significantly improved the accuracy of the AGB estimation performance.

**Table 6.** Fitting and testing statistics of the three datasets using RF model. Habitat: habitat dataset; RS: remote-sensing dataset; Combined: combined dataset; $R^2$: coefficient of determination; RMSE: root mean square error; ME: mean error; MRE: mean relative error; and MARE: mean absolute relative error.

| Species | Dataset | Fitting | | | Testing | | |
|---|---|---|---|---|---|---|---|
| | | $R^2$ | RMSE | NRMSE | ME (Mg/ha) | MRE (%) | MARE (%) |
| *Pinus yunnanensis* | Habitat | 0.2028 | 50.8261 | 0.4416 | 4.65 | 4.0639 | 38.2588 |
| | RS | 0.7074 | 30.7922 | 0.2675 | −0.3226 | −0.2819 | 34.8031 |
| | Combined | 0.7268 | 29.7535 | 0.2585 | 0.0963 | 0.0842 | 35.682 |
| *Pinus densata* | Habitat | 0.1903 | 63.5056 | 0.5322 | −13.776 | −12.797 | 45.9524 |
| | RS | 0.7511 | 35.2124 | 0.2951 | −9.4358 | −8.7654 | 30.5785 |
| | Combined | 0.7343 | 36.3738 | 0.3048 | −5.7433 | −5.3352 | 28.1384 |
| *Pinus kesiya* | Habitat | 0.7553 | 19.7669 | 0.1701 | 4.6127 | 3.9489 | 25.6779 |
| | RS | 0.4617 | 29.3169 | 0.2522 | 3.9373 | 3.3708 | 23.7645 |
| | Combined | 0.8316 | 16.3906 | 0.1410 | 3.7964 | 3.2502 | 25.3048 |

The maps of the predicted AGB for the three forests were generated using three datasets (habitat dataset, RS dataset, and combined dataset) under RF, as shown in Figure 7. For the *Pinus yunnanensis* forests and *Pinus densata* forests, the estimated AGB maps using the RS and the combined dataset were more heterogeneous than the estimated AGB maps using habitat datasets. However, for the *Pinus kesiya* forests, the heterogeneity of the estimated AGB map using the habitat dataset was higher than that of the other two datasets.

In order to further analyze the impact of a dataset on AGB estimation, the means and standard deviations of the residuals of test data under RF were calculated for the overall and different AGB segments, and the results are presented in Table 7. The AGB residuals of all predicted values showed similar trends. For instance, the means of the model residuals were highest in a case where only a habitat dataset was involved, followed by an RS dataset. Additionally, the means of the model residuals were lowest in a case where a combined dataset was involved, which was also closest to the measured value of the AGB. Furthermore, in cases where a combined dataset was used for AGB estimation of pine forests, the AGB standard deviation of the *Pinus densata* forests and *Pinus kesiya* forests was in the lowest grade, while that of the *Pinus yunnanensis* forests was in the middle grade. This result concludes that the prediction results of the model involving the combined dataset were more stable.

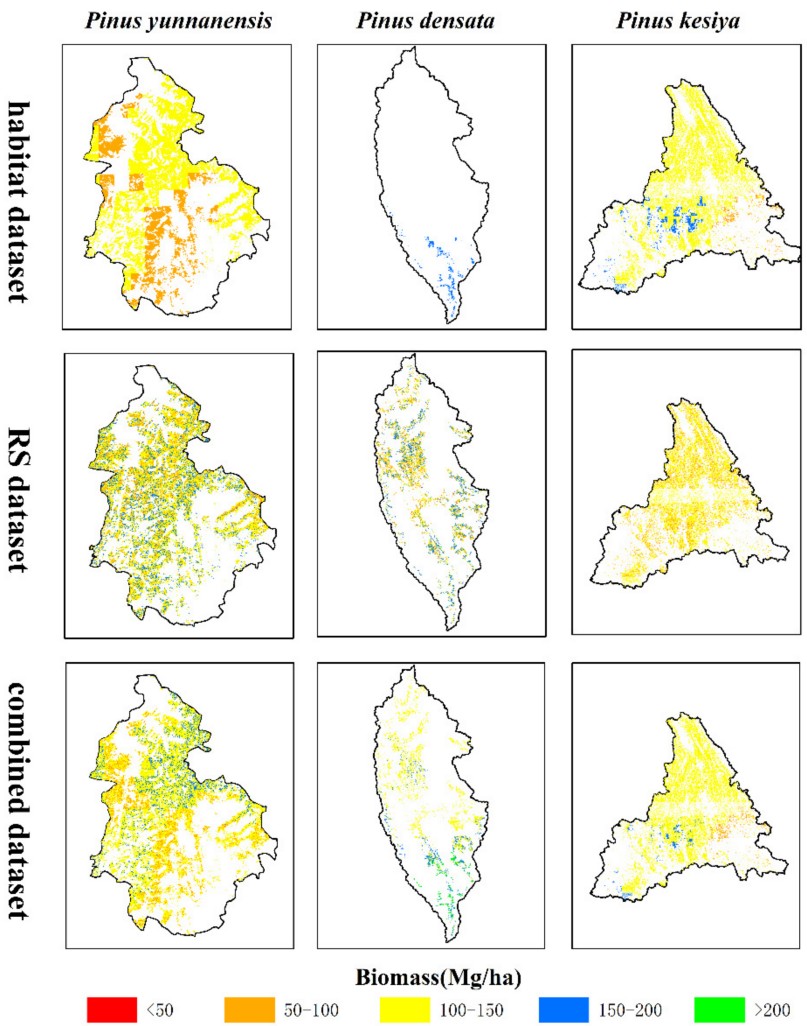

**Figure 7.** The spatial distributions of the predicted forest AGB values using the three datasets.

**Table 7.** Summary of the mean (μ) and standard deviation (σ) values of the residuals at different AGB classes for the three datasets based on the test dataset.

| Species | Dataset | <50 (Mg/ha) | | 50–100 (Mg/ha) | | 100–150 (Mg/ha) | | 150–200 (Mg/ha) | | >200 (Mg/ha) | | Overall | |
|---|---|---|---|---|---|---|---|---|---|---|---|---|---|
| | | μ | σ | μ | σ | μ | σ | μ | σ | μ | σ | μ | σ |
| *Pinus yunnanensis* | habitat | 51.08 | - - - | 34.29 | 19.46 | −13.41 | 31.10 | −49.54 | 12.97 | −128.17 | 0.67 | −4.65 | 50.52 |
| | RS | 43.89 | - - - | 41.74 | 30.38 | −10.79 | 25.18 | −39.83 | 29.24 | −128.59 | 27.03 | 0.32 | 53.68 |
| | combined | 40.67 | - - - | 40.00 | 28.85 | −10.07 | 30.36 | −41.96 | 19.46 | −122.51 | 2.59 | −0.11 | 51.86 |
| *Pinus densata* | habitat | 70.84 | 38.02 | 46.59 | 33.09 | −33.36 | 15.85 | 1.08 | 27.10 | −158.37 | 36.04 | 13.78 | 73.88 |
| | RS | 50.37 | 32.01 | 26.42 | 28.24 | −13.86 | 19.58 | −17.81 | 36.80 | −86.73 | 37.73 | 9.44 | 48.52 |
| | combined | 34.57 | 13.82 | 26.42 | 34.54 | −21.11 | 13.18 | −7.61 | 31.36 | −92.04 | 25.92 | 5.74 | 47.23 |
| *Pinus kesiya* | habitat | - - - | - - - | 11.71 | 11.99 | −2.33 | 32.63 | −61.54 | 6.84 | - - - | - - - | −4.61 | 33.19 |
| | RS | - - - | - - - | 26.34 | 15.86 | −14.74 | 16.23 | −56.95 | 20.45 | - - - | - - - | −3.94 | 32.79 |
| | combined | - - - | - - - | 12.65 | 12.25 | −3.68 | 34.32 | −53.54 | 9.46 | - - - | - - - | −3.80 | 32.56 |

On the basis of the statistical value of the predicted residuals for different AGB segments, in cases where the AGB was less than 100 Mg/ha, the predicted AGB values were all higher than the actual values. With the reduction in AGB values, the overestimation errors of the model tended to increase. However, when the AGB of pine forests exceeded 100 Mg/ha, the underestimation of the AGB prediction became increasingly clear. The estimated value of the AGB was more accurate in the range of 100 to 150 Mg/ha for the

*Pinus yunnanensis* forests and *Pinus kesiya* forests, while that of the *Pinus densata* forests was most accurate in the range of 150 to 200 Mg/ha.

In order to further analyze the impact of a dataset on AGB estimation for different AGB segments, the segmentation of *P. kesiya* forests with the smallest AGB span among the three forests was used as the standard to redefine the AGB segment, which was divided into < 100 Mg/ha, 100—150 Mg/ha, and > 150 Mg/ha. The means of residuals under RF at different AGB segments were calculated, and the result is shown in Figure 8. It is evident that the means of residuals using a combined dataset were lower than those using an optical remote-sensing dataset only in the two segments with larger AGB estimation errors (AGB < 100 Mg/ha or AGB > 150 Mg/ha). In the range of 150 to 200 Mg/ha, the same trend was revealed for the *Pinus yunnanensis* forests and *Pinus kesiya* forests, except for the *Pinus densata* forests. Based on this result, it is concluded that incorporating the habitat dataset into the optical RS dataset will reduce the number of estimation errors in cases with AGB < 100 Mg/ha or AGB > 150 Mg/ha compared to relying only on the Landsat 8 optical remote-sensing dataset.

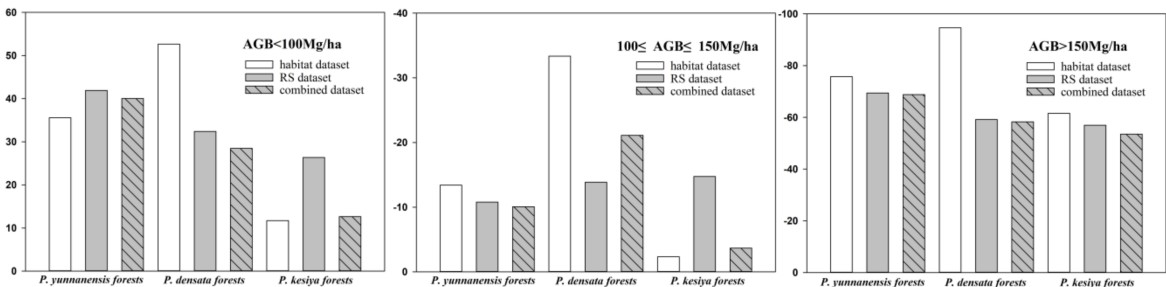

**Figure 8.** The means of residuals under RF at different AGB segments.

Table 8 indicates the top six most significant variables for AGB estimation by RF using a combined dataset. Thus, it also indicates the differences in significant variables for AGB estimation among tree species. The significant variables of AGB estimation of *Pinus yunnanensis* forests only came from the RS dataset; it mostly consisted of texture variables, and only one vegetation index was selected. For the *Pinus densata* forests, three important variables were obtained from the RS dataset, and the other three were obtained from the habitat dataset, which consisted of two temperature variables and one annual precipitation index. The habitat dataset was more important in the case of AGB estimation of the *Pinus kesiya* forests, since it required three precipitation indices, two seasonal variation coefficients of temperature, and one texture variable. Texture variables appeared in AGB estimation of both the *Pinus yunnanensis* forests and *Pinus kesiya* forests.

**Table 8.** Important variables for AGB estimation from different datasets.

| Species | Variables | $R^2$ |
|---|---|---|
| *Pinus yunnanensis* | B6_homo, B4_entro, B7_homo, SIPI, B4_semo, B7_diss | 0.7268 |
| *Pinus densata* | ARVI, SRI, EVI, bio4, bio7, bio12 | 0.7343 |
| *Pinus kesiya* | B4_mean, bio4, bio14, bio17, bio19, HSV | 0.8316 |

## 4. Discussion

### 4.1. The Selection of Modeling Algorithms

In this study, SLR was used to estimate the AGB of three pine forests with varying fitting datasets. The result showed that the estimation performance of AGB by SLR was not high, and the $R^2$ of more than 75% of them was below 0.3, and the highest $R^2$ was only 0.53. The mean NRMSE, ME, MRE, and MARE of the SLR model were larger than those of the other two non-parametric models, except for the MRE of SVM. Since a linear regression model can only analyze the relationship between a variable and independent variables from the linear point of view, it is difficult to obtain a favorable fitting performance by using a

linear model [76]. Although SLR can determine the important factors for regression through a significance test, it reduces the number of modeling variables and improves the estimation accuracy of the model only to a certain extent. However, due to the complex relationship between the variables and the AGB, it is difficult to use a linear regression model to explain such a complex nonlinear relationship, and for that purpose, non-parametric machine-learning algorithms, such as RF and SVM, are necessary [84]. In this study, the mean of $R^2$ using the non-parametric model for AGB estimation was all above 0.5, indicating a better fitting performance than the SLR. The fitting results for the two non-parametric models (RF and SVM), including the mean, maximum, and the standard deviation of $R^2$ and NRMSE, showed the estimation performance using RF was slightly better than that of SVM. Additionally, this trend was also reflected in MAE and MARE, which used absolute values as the measurement standard of test data errors.

The overall estimation performance of the AGB differed by about 5% between the two non-parametric models, although there was a certain gap in the estimation of AGB for different forests. In this study, the fitting performance of AGB estimation was measured by the mean value of $R^2$ of different fitting datasets of a certain forest under a specific estimation model. Under SLR, the result indicates that the cases with the highest $R^2$ of the AGB fitting estimation were the *Pinus kesiya* forests, *Pinus yunnanensis* forests, and *Pinus densata* forests, in descending order, which was inversely proportional to the sample sizes of AGB fitting. Additionally, the $R^2$ difference of this model among different forests was the highest among the three models. The $R^2$ of AGB estimation using RF were the *Pinus kesiya* forests, *Pinus densata* forests, and *Pinus yunnanensis* forests, in descending order. The AGB estimation of the *Pinus densata* forests with a larger data volume was significantly better, and the difference of $R^2$ among the varying forests was the lowest among the three models. This indicates that RF has a good interpretation and a strong degree of robustness for AGB estimations. This is consistent with the study of Zhang [85]. The descending order of $R^2$ using SVM was *Pinus kesiya* forests, *Pinus yunnanensis* forests, and *Pinus densata* forests, respectively, which was consistent with the SLR model. However, the variation coefficient of AGB estimation using SVM was significantly smaller than that of SLR. In other words, AGB estimation using SVM was less affected by the forest type than that of SLR. This indicates that SVM is better suited for small sample data as compared to SLR and RF, which is consistent with previous research conclusions [86,87].

### 4.2. Selection of Suitable Variables for AGB Modeling

In cases where a single dataset was used to estimate the AGB, estimation using an RS dataset performed better, except for the case of *Pinus yunnanensis* forests using a habitat dataset under SLR and *Pinus kesiya* forests using a habitat dataset under RF. This result indicates that in comparison with a habitat dataset, an RS dataset had excellent AGB estimation ability for a large area, which is consistent with previous studies [75,87,88].

In cases where an RS dataset was used for AGB estimation, texture variables were most frequently used in model construction, followed by the vegetation indices, which were obtained through ground object reflection data, as presented in Table 9. Therefore, texture information was of great significance for AGB estimations because texture information has the ability to describe the complex canopy structure of subtropical forest [61].

**Table 9.** Important variables in AGB estimation using remote-sensing dataset.

| Species | Model | Important Variables |
|---|---|---|
| *P. yunnanensis* | SLR | B7_homo |
| | RF | B6_homo, B4_entro, B7_homo |
| *P. densata* | SLR | SRI |
| | RF | ARVI, SRI, EVI |
| *P. kesiya* | SLR | B4_mean |
| | RF | B4_mean, B2_corr, B2_con |

*4.3. AGB Estimation by Incorporating the Habitat Dataset into the Models*

The AGB fitting performance conducted through combining a habitat dataset with a remote-sensing dataset was, to a certain extent, higher than that of the cases using a single dataset. Particularly in the SLR, the $R^2$ of AGB estimation using a combined dataset was more than twice as high as that of an RS dataset. In the non-parametric AGB estimation using a combined dataset, only the *Pinus densata* forests based on RF and *Pinus yunnanensis* forests based on SVM were slightly lower than the $R^2$ of an RS dataset, and the difference in $R^2$ was within 0.02. Recent studies have shown that the spatial pattern of AGB distribution of vegetation is consistent with the response of its habitat [89]. Integrating a habitat dataset representing environmental characteristics into an AGB estimation can compensate for an RS dataset's problem of low time span and improve the performance of AGB estimation. As early as 1996, Phinn et al. emphasized the importance of habitat to the ecosystems [90]. Habitat is the result of long-term development of vegetation, and accurate estimation of forest biomass requires knowledge of the characteristics of long-term forest development [91].

The employment of a habitat dataset not only improved the performance of AGB estimation on the whole but also reduced the number of overestimation errors in AGB estimation. The decreasing degree of overestimation errors was greatest in the case of *Pinus densata* forests, followed by *Pinus kesiya* forests, and finally, *Pinus yunnanensis* forests, respectively. As for the underestimation errors in AGB estimations, in comparison to the RF model using only the Landsat 8 OLI optical RS dataset, the combined dataset could reduce the underestimation errors across all segments of *Pinus yunnanensis* forests and *Pinus kesiya* forests with AGB greater than 100 Mg/ha, in addition to *Pinus densata* forests with AGB higher than 150 Mg/ha. A combined dataset can significantly improve the AGB estimations of pine forests [23], and a combined dataset is not necessarily the result of combining different remote-sensing datasets but also of combining a habitat dataset and an optical remote-sensing dataset. The combination of a habitat dataset and an optical remote-sensing dataset is more suitable for low AGB estimation (AGB < 100 Mg/ha) in cases where non-parametric modeling methods are used. Based on the fact that habitat has a profound impact on the richness and spatial distribution of species in a region [92], at least to a certain extent, the habitat dataset can represent the structural information of a forest in a region. Therefore, combining a habitat dataset with an RS dataset can compensate for the unreliability of an optical remote-sensing dataset for AGB estimation. In the case of AGB estimations of pine forests using a combined dataset, the habitat variables of *Pinus yunnanensis* forests did not rank among the top six most significant variables in the RF model, and the habitat variables of *Pinus densata* forests accounted for 50%, while those of *Pinus kesiya* forests accounted for 83%. Therefore, whether the estimated differences among the forests were related to the number of habitat variables that were included still needs further discussion.

*4.4. Comparison and Implication of Similar Studies*

In order to further analyze the research conclusions of this paper, we compared two papers that also applied Landsat images to estimate the AGB of *Pinus densata* forests in Shangri-la. Zhang et al. [33] used Landsat time series images and national forest survey data from 1987 to 2007 to produce parametric and non-parametric AGB estimations. In Zhang's study, i.e., AGB estimation without continuous image participation, the $R^2$ of SLR and RF were 0.46 and 0.87, respectively, and the MAE of the validation data were 20.48 and 22.47. The $R^2$ of the AGB estimation with the participation of 5-year sequence images were all increased to above 0.9, and the MAEs were reduced to below 10. In this study, the $R^2$s of AGB estimations that utilized a combined dataset were 0.17 and 0.73, and the MAEs were 58.82 and 34.68 under SLR and RF, respectively. The $R^2$ was lower than that in Zhang's study, but the result of the model fitting comparison was consistent. In other words, in the case of the *Pinus densata* forests, the AGB estimation of an RF model was more accurate than that of an SLR model. The estimation performance of this study was lower than that

of Zhang for the following reasons. First of all, Zhang not only used spectral variables and spectral conversion variables of RS data but also used terrain variables to reduce the impact of the slope on AGB estimation. The type of the forest ecosystem and the change in terrain both affect the estimation performance of AGB estimations [23]. However, this study only integrated the habitat dataset derived from bioclimatic data into the remote-sensing dataset for AGB estimation while ignoring the fact that the lack of topographic variables may reduce the estimation performance, especially in the case of the Shangri-La region with large terrain fluctuations. Secondly, the climate data from WorldClim used in this study were the mean values of various bioclimatic indicators obtained from 1970 to 2000. This time interval was significantly larger than the 5-year time interval estimated by the optimal AGB in Zhang's study. In addition to temporal resolution, the spatial resolution (1 km$^2$) of climate data was also significantly lower than that of the image data. Finally, the sample size of the *Pinus densata* forests used in this study (147 samples) was significantly larger than the 53 samples used by Zhang, which may have led to a lower estimation accuracy.

Ou et al. [30] incorporated age data as a dummy variable into a Landsat 8 optical image to estimate the AGB of *Pinus densata* forests. As a result, the RMSE of the linear regression dropped from 50.163 to 33.020, and the RMSE of RF dropped from 40.108 to 23.311. This proves that the age-fused optical RS combined dataset significantly improved the AGB estimation performance. It also indicates that the combined dataset that could improve the fitting performance of AGB estimations included the combination of different remote-sensing data sources [23,93], as well as the combination of remote-sensing data and non-remote-sensing data. For example, the combination of a habitat dataset and Landsat optical images in this study also significantly improved the overall estimation performance of the AGB and was able to reduce the number of overestimation and underestimation errors in cases where only an optical remote-sensing dataset was used for AGB estimation. Ou's study concluded that the estimation models that use age as a dummy variable perform best when AGB < 70 Mg/ha and AGB > 180 Mg/ha, while this study was the most accurate when the AGB was 150–200 Mg/ha, and the prediction accuracy was also slightly lower than Ou's study. However, in forests where the age of stands is difficult to obtain, such as pure forests with uneven ages and mixed forests with multiple dominant tree species, the habitat dataset used in this study can be regarded as an effective way to improve the performance of AGB estimation.

### 4.5. Limitation and Future Research

This study confirms that a nonlinear algorithm (RF and SVM) is more suitable for AGB estimation of the pine forests than SLR, and the integration of habitat information can improve the estimation accuracy of AGB estimation using Landsat optical images. However, to some extent, the following limitations still exist. Firstly, the three pine forests, which were selected for AGB estimation in this study, are located in different regions of the study area, and the environmental information, such as topography, climate, and physicochemical properties of the soil, varies. This difference will inevitably affect the habitat suitability for tree growth in the forest. Although none of the other factors play a significant role compared to the effects of climate [94], the inclusion of information on the elevation, land cover, and landscape spatial alignment, which have an impact on biodiversity [48,52,95], can reduce the uncertainty in habitat suitability calculations. Therefore, in later studies, we will try to combine more variables to improve the expressiveness of the habitat information.

Secondly, the data size of the estimated sample is not only related to the selection of the estimation model but also affects the estimation performance of the model. Due to the different distribution ranges of the three pine forests, there are differences in the number of trees obtained by the same sampling method. However, a sample size of 30 can meet the requirements for AGB estimation. Bao et al. emphasized that a minimum of 30 samples should be ensured in AGB estimation [96]. Rafaela et al. estimated the AGB of mangroves with a sample size of 30 and obtained a good estimation [97]. Moreover, in order to clarify the effect of sample size on AGB estimation using the combined dataset, we will select

different sample size data to study the AGB estimation of a specific pine forest in a certain area in the next step. Technology and means such as growth cones and LiDAR can also be used to obtain more samples to facilitate the smooth progress of research.

## 5. Conclusions

In this study, three common pine forests (*Pinus yunnanensis* forests, *Pinus densata* forests, and *Pinus kesiya* forests) in Yunnan Province, southwest China, were taken as examples for AGB estimation. The estimation was performed under a parametric model (SLR) and a non-parametric model (RF and SVM) based on a habitat dataset, Landsat8 OLI optical remote-sensing dataset, and a combined dataset produced by combining the two. The results indicate that (1) the non-parametric models of RF and SVM are capable of predicting the AGB of the three pine forests more accurately than the parametric model of SLR. RF is suitable for AGB estimation with a large sample size, while SVM is better suited for AGB estimation with a small sample size. (2) When a single dataset is employed for AGB estimation of the three pine forests, the resulting estimation performance of the habitat dataset is lower than that of the RS dataset, and the texture variables in the RS dataset are more significant in AGB estimation. (3) As compared to the overall fitting performance of the AGB estimations, which only use a single dataset, the combined dataset resulting from the combination of the habitat dataset and the RS dataset can improve the estimation performance to a certain extent. In particular, combining datasets can reduce the number of estimation errors in cases with AGB lower than 100 Mg/ha or exceeding 150 Mg/ha using RF.

**Author Contributions:** J.T. participated in the collection of the field data, conducted the data analysis, and wrote the draft of the paper; Y.L. (Ying Liu), L.L. and H.X. helped with the data analysis and writing of the paper; Y.L. (Yanfeng Liu) and Y.W. participated in the collection of the field data and data analysis; G.O. supervised and coordinated the research project, designed the experiment, and revised the paper. All authors have read and agreed to the published version of the manuscript.

**Funding:** This research was funded by the National Natural Science Foundation of China (grant numbers 31770677 and 31660202) and the Ten-Thousand Talents Program of Yunnan Province, China (YNWR-QNBJ-2018-184).

**Conflicts of Interest:** The authors declare no conflict of interest.

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
