# Peer review of "Enhancing Aboveground Biomass Estimation for Three Pinus Forests in Yunnan, SW China, Using Landsat 8"

_remotesensing, doi:10.3390/rs14184589_

Round 1

Reviewer 1 Report

 in this manuscript, a habitat dataset describing the distribution environment of forests, Landsat 8 OLI image data of spectral reflectance information, as well as a combination of the two datasets was employed to estimate the AGB of the three common pine forests (Pinus yunnanensis forests, Pinus densata forests, and Pinus kesiya forests) in Yunnan Province, the mancript is novel, however, the manuscript should be revised before being accepted.
(1) there are too few field plot, the much more sample plot should be added.
(2)the image of  the LC81320402016348LGN00, LC81320412016348LGN00 and LC81310412016325LGN00 have the over 50% cloud cover,  authors should replace them with  other remote sensing image.
(3) In 2.7 Model Evaluation section, I suggest Normalized RMSE should be used instead of RMSE.
(4) in Table 6. Statistics of AGB fitting results of different models, the Min. is 0.1559 for the best SVM, please give the data distribution of the result. e.g, box plot.

Author Response

Response to Reviewer 1 Comments

(x) I would not like to sign my review report

( ) I would like to sign my review report

English language and style

( ) Extensive editing of English language and style required

(x) Moderate English changes required

( ) English language and style are fine/minor spell check required

( ) I don't feel qualified to judge about the English language and style

Response: In the revised manuscript (MS), we read and re-edited the whole MS and corrected all the issues we found.

Yes

Can be improved

Must be improved

Not applicable

Does the introduction provide sufficient background and include all relevant references?

( )

(x)

( )

( )

Are all the cited references relevant to the research?

( )

(x)

( )

( )

Is the research design appropriate?

( )

(x)

( )

( )

Are the methods adequately described?

( )

(x)

( )

( )

Are the results clearly presented?

( )

(x)

( )

( )

Are the conclusions supported by the results?

( )

(x)

( )

( )

Response: In the revised MS, we took into account all the comments from the editor and three reviewers and made the corresponding changes. We revised all the sections of the MS and greatly improved the MS.

in this manuscript, a habitat dataset describing the distribution environment of forests, Landsat 8 OLI image data of spectral reflectance information, as well as a combination of the two datasets was employed to estimate the AGB of the three common pine forests (Pinus yunnanensis forests, Pinus densata forests, and Pinus kesiya forests) in Yunnan Province, the manuscript is novel, however, the manuscript should be revised before being accepted.

Response: In the revised MS, we took into account all the comments from the editor and three reviewers and made the corresponding changes. We revised all the sections of the MS.

  • there are too few field plot, the much more sample plot should be added.

Response: In the revised manuscript (MS), we compared with similar studies, demonstrated the feasibility of the 30 sample size on AGB estimation, and will further analyze the effect of sample size on the AGB estimation in the next research.

We added the section 4.5“Limitation and Future Research” as follows:

“the data size of the estimated sample is not only related to the selection of the estimation model, but also affects the estimation performance of the model. Due to the different distribution ranges of the three pine forests, there are differences in the number of trees obtained by the same sampling method. However, a sample size of 30 can meet the requirements for AGB estimation. Bao et al. emphasized that a minimum of 30 samples should be ensured in AGB estimation[98]. Rafaela et al. estimated the AGB of mangroves with a sample size of 30 and obtained good estimation results[99]. Moreover, in order to clarify the effect of sample size on the AGB estimation using the combined dataset, we will select different sample size data to study the AGB estimation of a specific pine forest in a certain area in the next step. Technology and means such as growth cones and LiDAR can also be used to obtain more samples to facilitate the smooth progress of research.”

  • the image of  the LC81320402016348LGN00, LC81320412016348LGN00 and LC81310412016325LGN00 have the over 50% cloud cover,  authors should replace them with  other remote sensing image.

Response: By reviewing the data, we have corrected the wrong part and revised the presentation of the table header, making it clear that the unit of cloud cover is percentage(%). LC81290452015052LGN00 has the largest cloud cover in the used images, and its cloud cover is 1.87%.

Table 2. The specific parameters of Landsat 8 OLI images.

Study Aera

Image ID

Average cloud cover(%)

Start time

Yongren

LC81300422016030LGN00

0.00

2016.01.30

Shangri-la

LC81310412016325LGN00

0.40

2016.11.20

LC81320402016348LGN00

0.73

2016.12.13

LC81320412016348LGN00

0.76

2016.12.13

Pu 'er

LC81290442015052LGN00

0.08

2015.02.21

LC81290452015052LGN00

1.87

2015.02.21

LC81310432015066LGN00

0.00

2015.03.07

LC81310442015066LGN00

0.00

2015.03.07

LC81300432015075LGN00

0.18

2015.03.16

LC81300442016046LGN00

0.00

2016.02.15

LC81300452016046LGN00

0.01

2016.02.15

LC81310452016069LGN00

0.41

2016.03.09

  • In 2.7 Model Evaluation section, I suggest Normalized RMSE should be used instead of RMSE.

Response: According to the suggestion, we added the model evaluation metric NRMSE in this study.

(4) in Table 6. Statistics of AGB fitting results of different models, the Min. is 0.1559 for the best SVM, please give the data distribution of the result. e.g, box plot.

Response: We added a boxplot to represent this issue at the reviewer's suggestion and added corresponding statements, as follows:

Figure 5. Box-plot of three algorithm for the AGB estimation of the Pinus forest

“To further analyze the impact of different algorithms on the AGB estimation, a box-plot of the three algorithm (SLR, RF, SVM) for the AGB estimation was made, as shown in Figure 5. It can be seen that the median fitting coefficients of SLR, RF and SVM are 0.1650, 0.7286 and 0.5361 respectively. RF is significantly higher than the other two models, and the interquartile rang (IQR) of RF is smaller than that of SVM. The median NRMSE of the three algorithms was 0.4350, 0.2634 and 0.2409 respectively. SLR having the largest error value and RF being slightly higher than SVM. The IQR of RF was the smallest among the three models and the RF model has the smallest estimation error dispersion. RF had the best performance of AGB estimation in the fitting data, followed by SVM and SLR ,indicating that the non-parametric model has better AGB estimation characteristics for pine forests.”

Reviewer 2 Report

The manuscript was well organized to show comparative analyses on above ground biomass estimation using Habitat dataset including HSV and 19 bioclimatic environmental variables, RS dataset including 56 texture variables and five commonly used vegetation indices, as well as Combined dataset by SLR, RF, and SVM with R2, RMSE, ME, MAE, and MARE for three Pinus forests. The manuscript fits the scope of this journal and should be published after revisions with minor comments below.

The unit of DBH and H as cm and m should be described in P7? The equation (4) might be divided by 1000000 instead of 1000 due to the unit of AGBs as g/m2 and the unit of AGBp as Mg/ha.

"Area Under the Curve (AUC)" should be described in P9.

Which dataset were used for Tables 6 and 7?

Was RF used for Table 9 like Table 8?

Band4, Band6, and Band7 should be B4, B6 and B7 in Table 11 like in other texts and tables.

Author Response

Response to Reviewer 2 Comments

(x) I would not like to sign my review report

( ) I would like to sign my review report

English language and style

( ) Extensive editing of English language and style required

( ) Moderate English changes required

( ) English language and style are fine/minor spell check required

(x) I don't feel qualified to judge about the English language and style

Yes

Can be improved

Must be improved

Not applicable

Does the introduction provide sufficient background and include all relevant references?

(x)

( )

( )

( )

Are all the cited references relevant to the research?

(x)

( )

( )

( )

Is the research design appropriate?

(x)

( )

( )

( )

Are the methods adequately described?

(x)

( )

( )

( )

Are the results clearly presented?

(x)

( )

( )

( )

Are the conclusions supported by the results?

(x)

( )

( )

( )

Response: In the revised MS, we took into account all the comments from the editor and three reviewers and made the corresponding changes. We revised all the sections of the MS and greatly improved the MS.

The manuscript was well organized to show comparative analyses on above ground biomass estimation using Habitat dataset including HSV and 19 bioclimatic environmental variables, RS dataset including 56 texture variables and five commonly used vegetation indices, as well as Combined dataset by SLR, RF, and SVM with R2, RMSE, ME, MAE, and MARE for three Pinus forests. The manuscript fits the scope of this journal and should be published after revisions with minor comments below.

Response: In the revised manuscript (MS), We read and re-edited the whole MS and corrected all the issues.

The unit of DBH and H as cm and m should be described in P7? The equation (4) might be divided by 1000000 instead of 1000 due to the unit of AGBs as g/m2 and the unit of AGBp as Mg/ha.

Response: According to the prompts, we add the unit of DBH and H in P7. And then we reviewed the literature and communicated with the builder of the formula in time, corrected the unit of the above-ground biomass of individual tree to be kilograms (kg), and added the source of the AGB model of individual tree. Secondly, as the units of sample plot AGB used in the subsequent modelling were ton per hectare (Mg/ha), a conversion of the sample plot biomass units was required, and the specific conversion formula is shown below.

"Area Under the Curve (AUC)" should be described in P9.

Response: According to the suggestion, we added the corresponding content as follows:

“The results showed that the habitat fitting AUC values of Pinus yunnanensis forest, Pinus densata forest and Pinus kesiya forest were 0.9889, 0.9906 and 0.9983, respectively. It is generally accepted the predicted result is accurate when the AUC value greater than 0.9 [75,76]. Habitat simulation AUC values for all the three pine forests were higher than 0.98. And the fitted AUC standard deviation of Yunnan pine forest and Pinus densata forest is 0.0018, and Pinus kesiya forest is 0.0004, all of which are less than 0.002, the accuracy is guaranteed.”

Which dataset were used for Tables 6 and 7?

Response: We re-wrote these sections according to the reviewer's comment and replace Table 6 with Figure 5 for more vivid expression as follows:

“AGB fitting was performed on the SLR, RF and SVM of the three pine forests using the habitat dataset, RS dataset and combined dataset, respectively. The R2 by the SLR model for the AGB estimation of Pinus yunnanensis forests ranged from 0.1039 to 0.2514, the R2 value of Pinus densata forests ranged from 0.0742 to 0.1650, and that of Pinus kesiya forests ranged from 0.0872 to 0.5331. The R2 of the SLR was primarily below 0.6. When an RF model was implemented, the resulting R2 of AGB fitting of Pinus yunnanensis forests ranged from 0.2028 to 0.7268, while Pinus densata forests ranged from 0.1903 to 0.7511, and Pinus kesiya forests ranged from 0.4617 to 0.8316. The R2 distribution was mostly higher than 0.7. The R2 of the SVM for the AGB fitting of the pine forest was mostly higher than 0.5. The R2 of the Pinus yunnanensis forests ranged from 0.1791 to 0.8100, the Pinus densata forests ranged from 0.1559 to 0.7285, and the Pinus kesiya forest was concentrated between 0.5034 to 0.7956.

To further analyze the impact of different algorithms on the AGB estimation, a box-plot of the three algorithm (SLR, RF, SVM) for the AGB estimation was made, as shown in Figure 5. It can be seen that the median fitting coefficients of SLR, RF and SVM are 0.1650, 0.7286 and 0.5361 respectively. RF is significantly higher than the other two models, and the interquartile rang (IQR) of RF is smaller than that of SVM. The median NRMSE of the three algorithms was 0.4350, 0.2634 and 0.2409 respectively. SLR having the largest error value and RF being slightly higher than SVM. The IQR of RF was the smallest among the three models and the RF model has the smallest estimation error dispersion. RF had the best performance of AGB estimation in the fitting data, followed by SVM and SLR ,indicating that the non-parametric model has better AGB estimation characteristics for pine forests.”

Figure 5. Box-plot of three algorithm for the AGB estimation of the Pinus forest

 “According to the results of a certain model, the dataset that provided the highest R2 in fitting data was applied to the AGB estimation of test data, that is, the combined dataset was used for all except for the RF estimation of the Pinus densata forests and the SVM estimation of the Pinus yunnanensis, which were RS dataset used. Thereby, the errors of the varying forests were determined. And the independence test index of the model (Table 5) was represented by the mean value of each individual model error measurement index.”

[Tip] Table 7 has been renamed to Table 5 due to the modification of MS

Was RF used for Table 9 like Table 8?

Response: Yes, the calculation results of table8 and table9 are obtained under the RF model. This had been specified at the beginning of Section 3.2.

“In order to explain how the AGB estimation of pine forests is affected by a dataset, an RF with a higher accuracy performance for an AGB estimation was selected to perform an AGB estimation of a habitat dataset, an RS dataset, and a combined dataset of them both. ”

Band4, Band6, and Band7 should be B4, B6 and B7 in Table 11 like in other texts and tables.

Response: We rewrote the relevant sections based on the comments of the reviewers.

[Tip] Table 11 has been renamed to Table 9 due to the modification of MS

Table 8. Important variables for AGB estimation from different datasets.

Species

Variables

R2

Pinus yunnanensis

B6_homo,B4_entro,B7_homo,SIPI,B4_semo,B7_diss

0.7268

Pinus densata

ARVI,SRI,EVI, bio4,bio7,bio12

0.7343

Pinus kesiya

B4_mean,bio4,bio14,bio17,bio19,HSV

0.8316

Table 9. Important variables in AGB estimation using remote sensing dataset.

Species

Model

Important Variables

P. yunnanensis

SLR

B7_homo

RF

B6_homo,B4_entro,B7_homo

P. densata

SLR

SRI

RF

ARVI,SRI,EVI

P. kesiya

SLR

B4_mean

RF

B4_mean,B2_corr,B2_con

Reviewer 3 Report

Introduction

- Although it is explained that forest maturity affects habitat, this study should support the fact that habitat affects the form of forest. Accordingly, it is necessary to more clearly modify the causal relationship of the paragraphs related to Habitat in the introduction section.

- "Random forest and support vector machines has two main shortcomings, ~ less clear than that of the parametric model." These two sentences seem essential to add the references.

Data, Study area

- It is necessary to add a sentence emphasizing the value of forests for the study area (necessity of AGB research).

- Numbers greater than or equal to thousands require commas, such as 1,000.

- Are Formula (1), (2), (3) all AGB_s?

- It is desirable to indicate the results of the pearson correlation analysis for the selection of Habitat factors.

Results

- In this study, did you estimate the biomass for the area corresponding to the training data (which is a part of the sample data) and validate it with the rest of the sample data (test data), or did you estimate the biomass for the entire study area with the model? If it is processed based on RS data, I am wondering why there is no AGB map for the entire study area in the study results.

Author Response

Response to Reviewer 3 Comments

(x) I would not like to sign my review report

( ) I would like to sign my review report

English language and style

( ) Extensive editing of English language and style required

( ) Moderate English changes required

( ) English language and style are fine/minor spell check required

(x) I don't feel qualified to judge about the English language and style

Yes

Can be improved

Must be improved

Not applicable

Does the introduction provide sufficient background and include all relevant references?

( )

( )

(x)

( )

Are all the cited references relevant to the research?

(x)

( )

( )

( )

Is the research design appropriate?

(x)

( )

( )

( )

Are the methods adequately described?

( )

(x)

( )

( )

Are the results clearly presented?

( )

(x)

( )

( )

Are the conclusions supported by the results?

( )

(x)

( )

( )

Response: In the revised MS, we took into account all the comments from the editor and three reviewers and made the corresponding changes. We revised all the sections of the MS and greatly improved the MS.

- Introduction

- Although it is explained that forest maturity affects habitat, this study should support the fact that habitat affects the form of forest. Accordingly, it is necessary to more clearly modify the causal relationship of the paragraphs related to Habitat in the introduction section.

Response: We added a description of the relationship between habitat and forest vertical structure in this section. As follows:

“This relationship exists due to the statistical association of habitat information with species abundance or probability of occurrence[49]. The abundance of species in a region is influenced by community structure at different scales[50], and forests are no exception. Studies have confirmed that the complexity of the vertical structure of forests is related to forest biodiversity[51,52]. Thus, the habitat information that can indirectly measure the biodiversity information in the region[53] can represent the vertical structure information of forest to a certain extent, especially on the large spatial scale[54].”

- "Random forest and support vector machines has two main shortcomings, ~ less clear than that of the parametric model." These two sentences seem essential to add the references.

Response: We added the references in this section.

“Although, the nonparametric model has two main shortcomings, the first is that it is sensitive to data[31,67], and second, its interpretation for the estimation process is less clear than that of the parametric model[6].”

Data, Study area

- It is necessary to add a sentence emphasizing the value of forests for the study area (necessity of AGB research).

Response: According to the suggestion, we added a description of the significance of AGB estimation in the study area.

“The pine forests in the province are mainly comprised of Pinus yunnanensis trees, Pinus densata trees, and Pinus kesiya trees[32,67,68]. The forests play an important role in ecological services and forest carbon sinks in the region[35,69], and forest biomass is the basis for forest carbon sink estimation [70].”

- Numbers greater than or equal to thousands require commas, such as 1,000.

Response: According to the suggestion, we modified the relevant content.

- Are Formula (1), (2), (3) all AGB_s?

Response: Formulas (1) (2) (3) are the above-ground biomass of individual tree for Pinus yunnanensis, Pinus densata and Pinus kesiya, and AGB_s in the formula has been replaced by AGBi.

- It is desirable to indicate the results of the pearson correlation analysis for the selection of Habitat factors.

Response: A radar plot representing the results of the Pearson correlation analysis of the habitat factors with the AGB had been added as recommended, and the results of the correlation analysis of the remote sensing factors had also been added together.

“The selected remote sensing variables and their correlation with AGB are shown in Figure 3. In the figure, the variables superscripted with "-" are factors that show a negative correlation with AGB.”

Figure 3. Radar-plot of remote-sensing data associated with forest AGB (a) Pinus yunnanensis forests; (b) Pinus densata forests; (c) Pinus kesiya forests.

“The selected habitat variables and their correlation with AGB are shown in Figure 4. In the figure, the variables superscripted with "-" are factors that show a negative correlation with AGB.”

Figure 4. Radar-plot of habitat data associated with forest AGB (a) Pinus yunnanensis forests; (b) Pinus densata forests; (c) Pinus kesiya forests.

Results

- In this study, did you estimate the biomass for the area corresponding to the training data (which is a part of the sample data) and validate it with the rest of the sample data (test data), or did you estimate the biomass for the entire study area with the model? If it is processed based on RS data, I am wondering why there is no AGB map for the entire study area in the study results.

Response: Based on the recommendation, we used RF that had the best AGB estimation performance in this paper, applied the model to different datasets for estimating the AGB of the three forests, and developed the AGB estimation map of the whole study area, as shown in Figure 7.

“The maps of the predicted AGB for the three forests were generated using three datasets(habitat dataset, RS dataset and combined dataset) under RF, as shown in Figure 7. For Pinus yunnanensis forests and Pinus densata forests, the estimated AGB maps using RS and combined dataset were more heterogeneous than that were in the estimated AGB maps using habitat datasets. But for Pinus kesiya forests, the heterogeneity of the estimated AGB map using the habitat dataset was higher than that of the other two datasets.”

Figure 7. The spatial distributions of the predicted forest AGB values using the three datasets.

Round 2

Reviewer 1 Report

I suggest that  the manuscript can be accepted in the current form.

Reviewer 3 Report

All comments were well reflected.  If it is not just conducted in a number of research areas, but if the reasons for the selection and comparative analysis are performed, it is likely that the paper will be of higher quality. However, I recommend that this manuscript to be published in Remote Sensing.